

# Relationship between variations of sea bottom temperature and American lobster catch rate off Southwestern Nova Scotia during 2008–2023

Adam M. Cook[1], Youyu Lu[1], Xianmin Hu[1], David Brickman[1], David Hebert[1], Chantelle Layton[1], Gilles Garric[2]

[1]Fisheries and Oceans Canada, Bedford Institute of Oceanography, 1 Challenger Dr.,  Dartmouth, NS, B2Y 4A2, Canada
[2]Mercator-Ocean International,  2 Av. de l'Aérodrome de Montaudran, 31400 Toulouse, France
*Correspondence to*: Youyu Lu (Youyu.Lu@dfo-mpo.gc.ca)

**Abstract.**    American lobsters (*Homarus americanus*) are an iconic species and are the socioeconomic and cultural mainstay for many communities across Nova Scotia. Describing the changes in population biomass and providing annual stock assessment advice for this species are required for sustainable fisheries. In many areas the best information available for providing this advice comes from commercial fisheries data. Often there is an assumed relationship between fisheries performance (catch per unit effort; CPUE) and available biomass, however several studies indicate that this relationship can be affected by external factors such as sea bottom temperature. Including bottom temperature when developing a standardized CPUE index will potentially address these concerns, however it has proven difficult in the past due to the lack of readily available (near real time) and unbiased bottom temperature data at the spatial and temporal scales required. Here we explore a global ocean reanalysis product of the European Union Copernicus Marine Service with an application to the fishery catch data from Lobster Fishing Area 33 during 2008-2023. Comparison with observational data shows that this reanalysis product provides realistic variations of sea bottom temperatures in this region. Next, a hierarchical generalized linear modelling approach is applied to evaluate the relationship between within-season changes in lobster CPUE and sea bottom temperature. Positive relationships between the rates of change of two model parameters, during the first 60 days of the fishing season (from mid-November to mid-January), are found in the majority of the 10 subregions. A standardized CPUE index with influence of bottom temperature included, compared to the index without such influence,  explains a high percentage of  the deviance of CPUE data and hence is more consistent with available stock biomass. The outcomes of the model evaluation and relationship analysis encourage  further applications of multi-decadal ocean reanalysis products to understand past changes, and the



development of ocean forecasts for predicting future changes in marine ecosystems and fisheries, a product with broad-reaching social-economical value.

## 1 Introduction

The Scotian Shelf (SS) is one of the regions of Atlantic Canada possessing highly valuable fishery resources. The fish population and fishery yields vary in both space and time, owing to multiple factors including changes in ocean conditions, which are known to affect fish bioenergetics and behaviour. For example, in the fishing areas of the snow crab (*Chionoectes opilio*) on the western SS, Zisserson and Cook (2017) identified the adverse impacts of a significant positive anomaly of sea bottom temperature in 2011/2012 on the local population of some life stages of snow crabs, with juvenile stages being the most affected as they are less likely to migrate out of suboptimal conditions. Similar studies have also been carried out for American lobster (*Homarus americanus*), another commercially important crustacean species in the region. American lobster are an iconic species and are the socioeconomic and cultural mainstay for many communities across Nova Scotia. Describing the changes in American lobster population biomass and providing annual stock assessment advice are required for sustainable fisheries. In many areas the best information available for providing this advice comes from commercial fisheries data. Often there is an assumed relationship between fisheries performance (catch per unit effort; CPUE) and available biomass, however several studies have shown that this relationship can be affected by external factors such as the sea bottom temperature (Wright and Liu 2024; Crossin et al 1998). In order to develop informative stock assessment advice incorporating bottom temperature two criteria must be met: 1) the bottom temperature information (from models or observations) should be available at appropriate spatial and temporal scales, unbiased (or able to be bias corrected) and available in real time (or approximately real time); and 2) the relationship between bottom temperature and CPUE should be evaluated outside the stock assessment model and follow expected trends.

In previous studies on the relationship between the fisheries performance and ocean conditions, sea bottom temperatures from both observation and numerical ocean models have been used. Ocean observations and models both possess strengths and weakness in depicting the complicated space-time variations of oceanic conditions of the SS, which are influenced by the strong multi-scale variability of atmospheric forcing at surface, and laterally from the Gulf of St. Lawrence, the Newfoundland Shelf, the Gulf of Stream and Labrador Current (e.g., Loder et al., 1998; Brickman et al., 2018). Observational data provide "ground truthing" of ocean variations and are very valuable for model evaluation and bias correction, but are often limited by sparseness in space and time. For example, the Atlantic Zonal Monitoring Program (AZMP) of Fisheries and Oceans Canada (DFO) conducts SS-wide surveys but these were mainly in July with some complementary seasonal surveys along specific sections (e.g., DFO. 2023; Hebert et al., 2024). The results of high-resolution ocean models fill gaps in observational data, but usually contain biases that need to be quantified and corrected. One systematic approach to reduce the model biases is the assimilation of observational data. This led to the creation of data assimilative ocean reanalysis products, such as product ref.





no. 2 (Table 1). Due to the open-ocean nature of most observational data used for assimilation, the accuracy of the reanalysis products for shelf and coastal seas needs to be evaluated using available shelf and coastal observational data.


We explore a global ocean reanalysis product of the European Union Copernicus Marine Service to 1) examine the relationship between lobster CPUE and sea bottom temperature on the SS and 2) develop a temperature corrected standardized index of CPUE including seasonal and interannual changes of bottom temperature. The analysis focuses on Lobster Fishing Area 33 (LFA33; Fig. 1a) due to the availability of lobster CPUE data from different subregions in this area during 2008-2023. The

sea bottom temperature is taken from product ref. no. 2 because of its high resolution and continuous coverage in space and time. This allows examining the variations of the relationship among different subregions, not previously detailed in studies. As discussed above, prior to this relationship analysis, the accuracy of product ref. no. 2 needs to be first evaluated with available observational data. Positive outcomes of the model evaluation and relationship analysis could encourage further application of multi-decadal ocean reanalysis products to reveal more linkages between variations in ocean conditions and

marine ecosystems and fisheries. This could further encourage the development and improvement of ocean forecasts for predicting future changes in marine ecosystems and fisheries, a product with broad-reaching social-economical value.

## 2 Datasets

A number of datasets were used in this study. Commercial fishing logbooks (hereafter "Logbooks") provide the most detailed accounting of the spatial and temporal extent of effort and landings for commercial lobster fishing currently available. These

logbooks are completed daily and submitted monthly to Dockside Monitoring Companies for data entry and transmission to the Commercial Data Division of DFO. Logbooks contain fishing information including Date Fished, Vessel Registration Number, Fishing Licence Number and Reporting Grid Number(s). For each day fished, Grid Number, Effort (Number of Traps Hauled) and Estimated Landings are provided. Based on the data in logbooks, we derive product ref. no. 1 (Table 1) , which is the estimates of daily CPUE (kilograms per trap haul) for 10 different subregions of LFA33 (Fig. 1a).


Daily sea bottom temperature on the Scotian Shelf, from 1 January 1993 to 31 December 2023 are obtained from product ref. no. 2, which is a global ocean reanalysis available on a horizontal grid of 1/12° in longitude/latitude and 50 vertical levels. Major ocean observational datasets are being assimilated to generate this data product (hereafter GLORYS).

The observed bottom temperature is obtained from two products. Product ref. no. 3 is the July AZMP survey. Product ref. no. 4 is a geo- and date- referenced dataset of observed bottom temperatures collected both systematically on DFO surveys and opportunistically through collaborations with the fishing industry (Fig. 1a; red solid circles; hereafter DFO Temp database).






| Product Ref. No. & Abbreviation | Product ID & type | Data Access | Documentation |
|---|---|---|---|
| 1: Logbooks | Estimates of daily catch per unit effort (CPUE; kilograms per trap haul) for 10 subregions of the Lobster Fishing Area 33 of east coast of Canada, based on commercial fishing logbooks | Commercial Data Division, Maritimes Region, DFO | Cook et al., 2020 |
| 2: GLORYS | GLOBAL_MULTIYEAR_PHY_001 _030, numerical models | EU Copernicus Marine Service Product (2023)  https://doi.org/10.48670/moi -00021 | Product User Manual (PUM): Drévillon et al., 2023a Quality Information Document (QUID): Drévillon et al., 2023b Journal article: Lellouche et al., 2021 |
| 3: AZMP | Bottom temperature from the AZMP July survey | https://www.dfo-mpo.gc.ca/science/data-donnees/azmp-pmza/index-eng.html | DFO, 2023; Hebert et al., 2024. |
| 4: DFO Temp Database | A database of geo- and date-referenced observed bottom temperatures collected both systematically on DFO surveys and opportunistically through collaborations with the fishing industry | Population Ecology Division, Maritimes Region, DFO | |

**Table 1: Data product reference table**





**95**

**Figure 1: (a)** The division of 10 subregions of LFA33 off Southwestern Nova Scotia, the locations of observed bottom temperatures during lobster fishing season of 2008–2023 (red solid circles), and the bathymetry contours of 50, 100 and 500 m. **(b)** Violin plots of observed bottom temperature during the commercial lobster fishing season (product ref. no 4) minus that extracted from reanalysis (product ref. no. 2) at matching times and grid points nearest to the observations, for subregions 1 to 10. **(3)** Daily time series of bottom temperature from product ref. no. 2 (black curve) and the lobster CPUE (product ref. no. 1, magenta solid circles) during lobster season of 2008-2023 in subregion 5.

## 3 Analysis methods and results

### 3.1 Evaluation of modelled bottom temperature with observational data

Figure 2 presents the spatial distribution of bottom temperature anomalies in July from the AZMP survey (product ref. no. 3, with certain level of horizontal interpolation being applied) and GLORYS reanalysis (product ref. no. 2), for five selected years 2008, 2012, 2015, 2022 and 2023. Focusing on the SS (east of 66ºW), the spatial patterns and the magnitudes of





anomalies during these years agree well. Among the five years, 2008 shows cold anomalies from the central deep basin (Emerald Basin) to the offshore bank (Western/Emerald Bank) and near the shelf break, with near normal conditions along the coast. In 2012, significant warm anomalies occurred along the coast and the western part of the SS (up to 3ºC), and over the offshore bank; and near-normal condition in the central basin. In 2015 and 2022, strong warm anomalies occurred in the eastern and western SS, respectively. In 2023, observations suggest moderate warm anomalies (about 1ºC) over two large areas in the eastern and western SS, while the reanalysis over-estimated the magnitudes of temperature anomalies in the eastern SS. On the SS, similar agreement and disagreement are found for other years during 1993-2023 (figures not shown).

The ocean model used to create the reanalysis does not include tides so the agreement between the observational and reanalysis data on SS is possible due to the generally weak tidal mixing in the region. In the Bay of Fundy (west of the SS) tides and tidal mixing are strong and the reanalysis data shows larger discrepancy with observations, e.g., in 2022 and 2023. The differences in the impacts of tidal mixing on ocean temperatures, between the SS and Bay of Fundy, have been quantified using model sensitivity experiments by Wang et al. (2020).

Figure 3 presents the monthly time series of bottom temperature during 1993-2023 from the GLORYS reanalysis and the July values from AZMP observations during 1993-2023 for the ten subregions for LFA33. Note that in regions 1-3 the observation are very limited and the observed July values are heavily influenced by the spatial interpolation. Based on reanalysis data, the seasonal variations are strong in subregions 1, 2 and 3, along the coast with maximum water depth of about 50 m. In these three subregions, depth-time sections of water temperature (figures not shown) demonstrate that 1) for January-March the whole water column is well mixed and the impacts of surface cooling reach the bottom, resulting in a minimum bottom temperature near 0ºC; 2) in April-August the impacts of surface warming penetrate downward due to vertical diffusion, leading to a steady increase of bottom temperature; and 3) in September-October the ocean's surface heat loss results in strong mixing (convection) of the heat in upper layer through the whole water column, resulting in a maximum bottom temperature of about 10ºC. The seasonal variations of bottom temperature are relatively weak in subregions 4-10 with water depths reaching ~200 m. In these regions, depth-time sections of water temperature (figures not shown) show that in winter the impacts of surface cooling do not reach the bottom, and in summer and fall there exists a "Cold Intermediate Layer" at depths of 50-100 m. This results in a weak fluctuation of monthly bottom temperature around 5ºC throughout of the year.

For all the ten subregions, the reanalysis data shows interannual variation of bottom temperature, accompanied by stronger or weaker seasonal variations. The interannual variations of July bottom temperatures, from the GLORYS reanalysis and AZMP observations, show high correlations (0.68-0.92) in subregions 4-10 with weaker seasonal variations. In subregions 1-3 with stronger seasonal variations, the correlation values (0.53-0.67) are relatively low but are still statistically significant. It is also notable that the reanalysis overestimates the observed July bottom temperature in subregions 1-3, while slightly underestimates or obtains similar values as the observations in subregions 4-10.



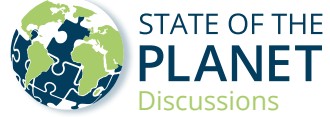

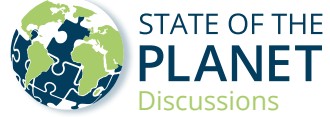





**Figure 2: Spatial distributions of bottom temperature anomalies (°C) in July of 2008, 2012, 2015, 2022 and 2023, from (left) AZMP observations (product ref. no. 3) and (right) GLORYS reanalysis (product ref. no. 2) for regions with water depth less than 500 m. Bathymetry contours of 131 m and 500 m are overlaid. "BoF", "EB" and "WB" in top-left panel denote the locations of Bay of Fundy, Emerald Basin, and the Western/Emerald Bank, respectively. The temperature anomalies are referenced to the July average during 1991-2020 and 1993-2020 for data show on left and right columns, respectively.**



**Figure 3: Bottom temperatures averaged for each of the 10 subregions of LFA33: (red curves) monthly time from reanalysis (product ref. no. 2) and (solid red circle) July values from observations (product ref. no. 3). The correlation coefficients between the modelled and observed July values are indicated by the *r* values in each panel.**



Figure 1b compares the observed bottom temperature during the lobster fishing season (product ref. no. 4) and the corresponding values from the GLORYS reanalysis in the form of "violin plots". The observations are matched to the GLORYS data values by date and nearest neighbour distances, across each of the subregions within LFA 33, during the lobster fishing season (December-May). The matching requires the distance between an observation location and reanalysis grid to
be less than 5 km. The overall median difference between observations (n=28794) and the reanalysis is 0.13 °C. Bottom temperatures in the westernmost, inshore and mid-shore subregions (1, 2, 5 and 6) are lower in the observational data than the reanalysis (median difference -0.26 °C), whereas in the offshore and eastern subregions (3, 4, 7, 8, 9 and 10) the observational data is higher (median difference 0.18 °C). Although the median differences are quite low, point differences ranged between +/- 10 °C.


In summary, evaluations with available observational data suggest that space-time variations of bottom temperatures from GLORYS on the SS are quite realistic, and are hence used to analyze on the relationship between variations of bottom temperature and lobster CPUE in LFA33.

### 3.2 Relationship between variations of bottom temperature and lobster catch rate off southwestern Nova Scotia

Two separate analyses are performed to explore 1) the relationship between the temperature changes within the fishing season and lobster CPUE and 2) the development of standardized CPUE indices with and without annual and seasonal bottom temperature corrections on the overall catch rate trends.

The first analysis focuses on the first 60 days of the fishing season (from mid-November to mid-January). As lobsters moult and grow during summer months, the lobster fishery in this area is considered as a recruitment fishery (i.e., a large component of the fishery are lobsters having reaching minimum legal size that year). As an example, Fig. 1c shows the time series of the
lobster CPUE and bottom temperature in subregion 5 during 2008-2023. Prior to the fishing season the fishable biomass can be considered as an unfished state and sea bottom temperatures remain high (from seasonal increases in spring and summer), hence the CPUE is maximized at the start of each fishing season. The CPUE rapidly decreases over the course of the first half of the fishing season (till mid-January) due mainly to depletion (cumulative removal of individuals through harvesting) and
also possibly the cooling of bottom water.

Conversely, the decreasing rates of both CPUE and bottom temperature have interannual variations. The rate of CPUE change is assumed to be independent of the initial size of the harvestable lobster stock. Its relationship with the bottom temperature, if identified, indicates the impacts the interannual variations of bottom temperature on the fishery performance. Here we adopt a hierarchical generalized linear modelling (HGLM; Lee and Nelder, 1996) approach to explore such relationship. In our
analysis, different quantifications of the bottom temperature, e.g., the mean, median and rates of changes, have been tried. The



highest correlation is found between the rates of change of bottom temperature and CPUE, and the HGLM analysis between them is formulated as

$$C_i = \beta_0 + \alpha_{j[i]} + (\beta_1 + \gamma_{j[i]})ST_i + \varepsilon_i$$
$$\alpha_j \sim N(0, \sigma_{p,0}^2)$$
$$\gamma_j \sim N(0, \sigma_{p,1}^2)$$


For each subregion ($j$) and each year ($i$), the rates of changes of CPUE and bottom temperature are denoted as $C_i$ and $ST_i$, respectively. The HGLM describes their relationship in each subregion and takes account of the random spatial effects on both the intercept $\beta_0$ and the slope $\beta_1$, denoted as $\alpha_{j[i]}$ and $\gamma_{j[i]}$ which are specified as a normal distributions ($N$) with mean 0 and variance $\sigma_{p,0}^2$ and $\sigma_{p,1}^2$ respectively.


The daily values of bottom temperature from GLORYS, averaged for each subregion, are obtained, then the rate of change ($ST_i$) is estimated through linear regression. For the CPUE, the logbook data can be variable in both the number of observations and scale of fishing, with time and in different subregions. Hence, in each subregion the rate of change of CPUE ($C_i$) over the first 60 days of each fishing season is estimated using weighted robust linear regression, with the total trap hauls being used

as the weighting variable.

Positive relationships between $C_i$ and $ST_i$, are found in the majority of the subregions of FLA33 (Fig. 4). The annual values of $C_i$ and $ST_i$ are within confidence intervals of the HGLM-derived relationship obtained in subregions 1, 2, 3 and 6. This positive relationship is less evident in subregions 4 and 8, where the CPUE data do not contain clearly defined depth ranges,

hence fishing likely occurred in smaller areas which do not represent the geographic extent of these subregions.

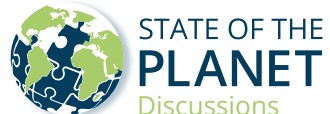

**Figure 4. The slope of lobster CPUE (CPUE/day, y-axis) versus slope of bottom temperature from GLORYS ( ºC/day, x-axis) during the first 60 days of the fishing season in subregions 1-10 of LFA33. Dots represent data from different years, and the red lines are the relationship obtained by applying the HGLM approach, with the shading denoting the confidence intervals.**





In the second analysis, two generalized additive models (GAM; Wood 2017) for standardizing lobster CPUE are developed,

one as the 'base-line' without incorporating bottom temperature and the other as the "temperature corrected", denoted as CM and CMT respectively. GAMs are chosen as they do not force a functional shape to the relationships (i.e. linear, quadratic, etc.) between dependent and independent variables rather they use penalized splines (denoted as *s*) to allow flexible relationships. The CM model is

$$CPUE_{ijk} = \beta_0 + s(D_{ijk}) + \beta_1 Y_{ij} + \varepsilon_{ijk}$$

And the CMT model is

$$CPUE_{ijk} = \beta_0 + s(D_{ijk}) + s(T_{ijk}) + \beta_1 Y_{ij} + \varepsilon_{ijk}$$

where subscripts *i* and *j* represent the year and sub-region, and k represents the day of fishing season. In the above equations, $D_{ijk}$ denotes the date corresponding to the day of the fishing season which varies with *i*, *j* and *k*; $Y_{ij}$ represents the Year variable that varies with *i* and *j* only; $\varepsilon_{ijk}$ represents a random error term; and $T_{ijk}$ is the bottom temperature included in the CMT

model. We focus on the conditional predictions from both models for a fixed day 10 of the fishing season (when CPUE is the highest). For CMT, $T_{ijk}$ is replaced by the catch-adjusted mean temperature across all years and all subregions on day 10 of the fishing season, calculated as :

$$CA_{10} = \frac{\sum(C_{ij10} \times T_{ijk10})}{\sum C_{ij10}}$$

where $C_{ij10}$ is the CPUE for day 10. The catch-adjusted mean temperature is used to account for the uneven distribution of the

fishing effort and catch, thus to ensure that the predictions from CMT with the impacts of bottom temperature included can be meaningfully compared with the predictions of CM.

Figure 5 compares the annual time series of the standardized CPUE indices from CM and CMT on day 10 of the fishing season averaged for all the 10 subregions of LFA33. The deviance of CPUE data explained by CMT is 59.6%, 5.8% higher than

53.8% by CM. Evident differences between the CM and CMT annual CPUE indices result from differences in bottom temperature: CPUE estimates from CM are higher than those from CMT in years when bottom temperatures are higher. Essentially, the temperature standardization accounts for the effects of warmer or cooler waters on the catch rates, and thus improves the relationship between CPUE and fishable biomass.





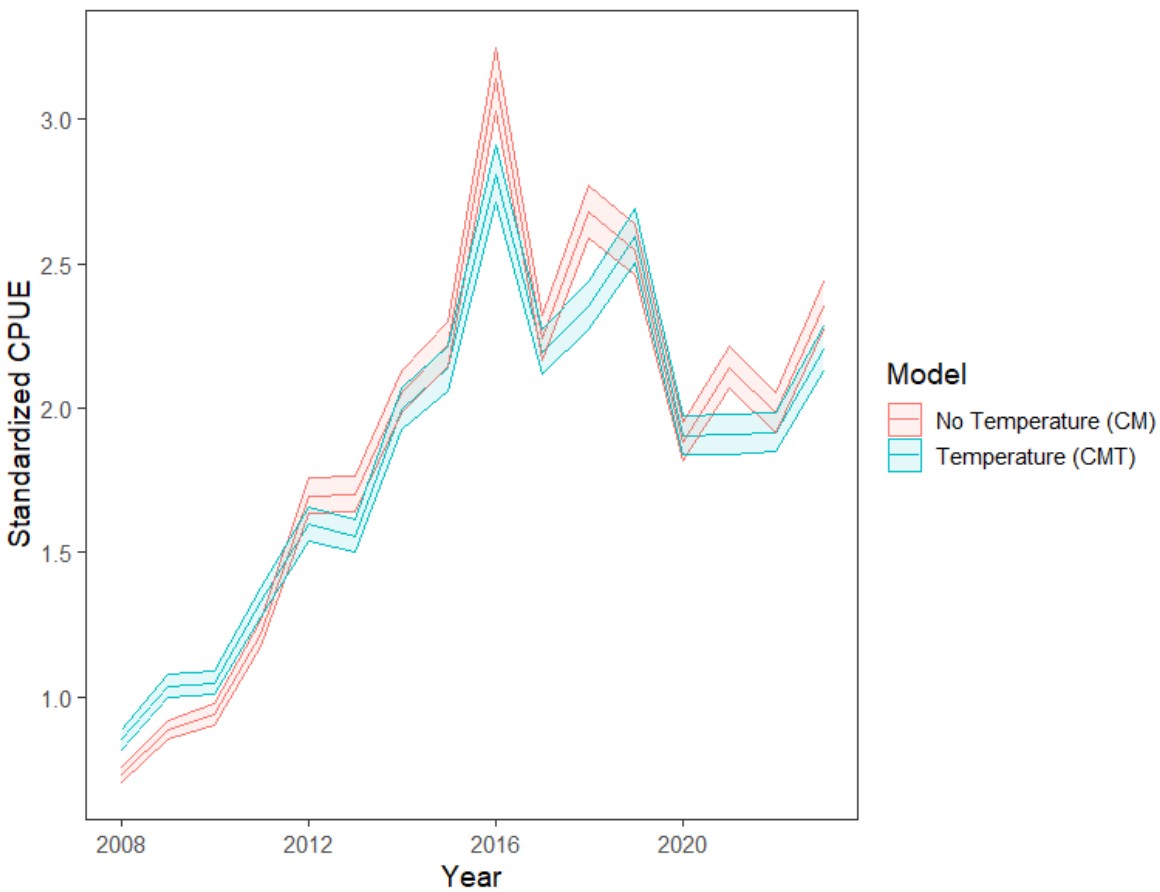

**Figure 5: Modeled index of annual lobster CPUE for LFA 33 with (blue) and without (red) bottom temperature as a predictor variable. Solid line represents mean effect whereas shaded polygons represent the 95% confidence bounds.**

**4 Conclusions and discussions**

The state-of-the-art ocean reanalysis (product ref. no. 2) is seen to provide the information required to develop better models describing changes in fishable biomass for American lobster stocks which will reduce uncertainty in fisheries advice and potentially lead to more sustainable fisheries. This reanalysis product meets the criteria of being unbiased, available in near real-time, with sufficient spatial and temporal resolution. Due to complex regional oceanographic processes under the influence of various forcing factors, observational data are insufficient to fully quantify the space-time variations of ocean conditions on the Scotian Shelf, and are even more sparse during the lobster fishing season from late autumn to early spring. In this study, evaluation shows that this data assimilative reanalysis is able to reproduce the features of variations derived from available observations during 1993-2023, on the Scotian Shelf where the influences of tidal mixing are relatively weak. The agreement



is less satisfactory in the adjacent Bay of Fundy where tidal mixing is strong while the ocean model used to generate the reanalysis does not include tidal mixing.

The evaluation results encourage the application of the reanalysis product to explore the relationship between the variations of
sea bottom temperature and lobster catch rate within LFA33. Both lobster CPUE and bottom temperatures have strong seasonal variations, the rapid decrease of CPUE during the first half of the season (mid-November to mid-January) can be partially described by the decreasing bottom temperature. Focusing on the CPUE changes during the initial 60 days of the fishing season (independent of stock size), a hierarchical generalized linear modelling approach (accounting for random spatial effects) is applied to explore the relationship between the rate changes of the CPUE and the rate of changes of the bottom temperature in
the 10 subregions. Clear positive relationship between the two rates are found in the majority of the subregions. This relationship is consistent with the previous understanding that cold bottom temperature anomalies reduce the movement of lobsters and hence the catch rate. The relationship is strong in subregions 1, 2, 3 and 6, but is less tight in subregions 4 and 8 which can partially be attributed to issues in the lobster CPUE data. This result is congruent with previous reports which indicate that sea temperatures can play a role in fisheries performance (e.g., Wright and Lio 2024). Incorporating bottom
temperatures, the estimated values of CPUE indices will decrease in warm years and increase in cool years, thereby reducing some of the uncertainty in the relationship between CPUE and fishable biomass. This is a valuable contribution to the development of stock assessment advice and an improved understanding of the dynamics of lobster populations.

Other quantitative measures of the lobster CPUE data (i.e., in different stages of the lobster fishing season) and their
relationship with variations of sea bottom temperature will be further explored in ongoing studies. The outcomes of this model evaluation and relationship analysis encourage further application of multi-decadal ocean reanalysis products to study changes in the marine ecosystems and fisheries. Furthermore, this also encourages the ongoing development and improvement of ocean forecasts for potentially predicting future changes in marine ecosystems and fisheries, a product with broad-reaching social-economical value.

**Data and code availability**

The data used in this study are available as described in Table 1. The code used in this study can be accessed via a gitlab repository upon request via email to the corresponding author.

**Author contribution**

AMC and YL led the conceptualization of the study and the writing of the manuscript. AMC contributed to the collection and
compiling of lobster data and bottom temperature data during the lobster fishing season, the analysis on relationship between

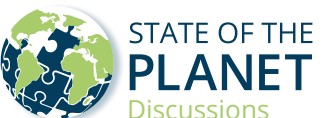

the lobster catch rate and bottom temperature, and the writing of the above aspects. XH carried out evaluation of GLORYS12v1 with the AZMP survey data. DH and CL provided the AZMP data. All authors contributed to conceptualization of the study, and editing and reviewing the manuscript.

**Competing interests**

The authors declare that they have no conflict of interest.

**Acknowledgements**

We appreciate DFO and Mercator-Ocean International for supporting the scientific exchanges and collaboration between the staff of both organizations, in recent years under a collaborative agreement. Dr. Karina Von Schuckmann provided insightful comments and advices in developing this manuscript, Drs. Nancy Soontiens and Ben Zisserson served as internal reviewers

for the manuscript.

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
