# Peer review of "Relationship between variations of sea bottom temperature and American lobster catch rate off Southwestern Nova Scotia during 2008–2023"

_State of the Planet, 2024_

## Author Response (AR2)

Reviewer 1

*This study has two aims, to compare observational ocean bottom temperature data with reanalysis data from a global model, and to develop a standardised CPUE index which accounts for temperature. Rigorous treatment is given to validating the use of the reanalysis product. Being able to use these data and not solely rely on observational data is very beneficial. I believe this makes a valuable contribution to the scientific literature and can pave the way for further research. I am relatively new in my career studying this fishery, so my comments are really more questions. I apologise if I've misinterpreted anything.*

Response reviewer 1:

Thank you for your thoughtful review and recommendations. Several of the comments were related and thus responses will be collapsed where appropriate.

Comment 1:

*Some more explanation on how the CPUE index might be used to inform/adjust stock assessments might be helpful. Is the hope for this to be used for official stock assessments or maybe used alongside traditional methods?*

*Is it possible that some of the increase in CPUE over the time period is due to increased vessel efficiency? Although not the focus of the paper, it can be acknowledged that there are still limitations to using CPUE indices as a proxy for biomass.*

In lines 37-45 of the introduction we provide a brief outline of the stock assessment approach used along with some of the caveats. We will provide more information in this section, as the reviewer's comment suggested we have not laid out the information clearly enough. We will also include a few sentences in the discussion on the caveats that remain with CPUE, even after standardizing for temperature.

Comment 2:

*A bit more information could be given that describes the physiological effect of temperature on lobster, and why this change in abundance is evident. Is this mostly because of lobster being more active when temperatures are warmer and more susceptible to being caught?*

We have eluded to the impact of temperature on lobster in several places in the document, however we will include a section in the introduction (Line 43) which

describes the physiological and behavioral effects temperature has on lobster and how this influences the interaction with the baited trap fishery.

Comment 3:

*There are some differences between the observational data and the reanalysis product in certain areas, is it possible to say how this might alter the calculated CPUE indices if it were to use observational data instead?*

The observational data were typically too sparse to be used to develop a CPUE index which is why we have explored the applicability of the GLORYSv12 product. Direct comparisons between the observations and GLORYSv12 indicated negligible biases (Figure 1b) and therefore we do not suspect any benefit in developing an observational CPUE index. No action taken.

Comment 4:

*Figure 5 shows the CPUE indices for all of LFA 33, would it also be helpful to show them for each of the 10 regions?*

The stock advice for LFA 33 is for the 10 regions combined and would be what is shown in Figure 5. We do not feel the addition of an extra figure to show the relationships separated by region would add to the overall story of the paper. No action taken.

Comment 5

*Besides these questions, I would suggest a general clean-up of the formatting, paragraph structure, and proofreading. Thank you for the opportunity to review this paper*

Thank you for the comment. We will go through the paper and clean up editorial and complete a thorough proof reading. Track changes in the document outline all the editorial corrections made.

*Reviewer 2:*

*The paper examines the relationship between lobster CPUE and sea bottom temperature on the Scotian Shelf in Lobster Fishing Area 33 during 2008-2023 and develops a temperature corrected standardized index of CPUE including seasonal and interannual changes of bottom temperature. For the bottom temperature a global ocean reanalysis product of the EU Copernicus Marine Service was used, validated with observational data for the area prior to its application in the study. The paper is concise, very well written and structured and easy to follow. The presented methodology is well formulated and robust and the conclusions drawn on the basis of obtained results are appropriate.*

*English is also very good. I only spotted one small error in line 51: Replace 'Gulf of Stream' with 'Gulf Stream'. I read the manuscript with pleasure and I would like to congratulate the authors on this study and its presentation in the manuscript. I highly welcome the study that brings together oceanographic and fisheries data and I recommend this paper for publication. If there is anything that authors could add, it would be a paragraph elaborating on preferred lobster habitat and how temperature affects its physiology.*

Thank you for your comments. We will make the editorial adjustment and will add a section on the lobster thermal physiology and how that impacts CPUE (comment 2; reviewer 1).